# Neuroanatomy of the Cetacean Sensory Systems

**DOI:** 10.3390/ani14010066

**Published:** 2023-12-23

**Authors:** Steffen De Vreese, Ksenia Orekhova, Maria Morell, Tommaso Gerussi, Jean-Marie Graïc

**Affiliations:** 1Laboratory of Applied Bioacoustics (LAB), Universitat Politècnica de Catalunya-BarcelonaTech (UPC), 08800 Vilanova i la Geltrú, Spain; 2Department of Comparative Biomedicine and Food Science (BCA), University of Padova, 35020 Legnaro, Italy; ksenia.orekhova@phd.unipd.it (K.O.); tommaso.gerussi@studenti.unipd.it (T.G.); jeanmarie.graic@unipd.it (J.-M.G.); 3Institute for Terrestrial and Aquatic Wildlife Research (ITAW), University of Veterinary Medicine Hannover, Foundation, 25761 Büsum, Germany; maria.morell@tiho-hannover.de

**Keywords:** senses, nervous system, dolphins, whales, vision, hearing, somatosensory, gustation, olfaction

## Abstract

**Simple Summary:**

Cetaceans, which include dolphins, porpoises, and whales, show astounding and distinctly specialised adaptations of their senses that allow them to thrive in the aquatic environment. They possess the abilities to echolocate, are likely to discern bioelectric fields, sense water currents, and perform various other sensory tasks, amidst shifting and challenging environmental circumstances. For many of their senses, we still lack a basic understanding of their underlying physiology. In this review, we provide a summary of the neuroanatomical knowledge of cetacean sensory systems to date. We describe the peripheral sensory cells in organs like the eye, ear, and skin, and trace the nervous pathways up to the brain where the signals are processed in dedicated areas. We highlight recent advances and knowledge gaps that exist at the various anatomical levels, accentuating the pressing need for further research to bridge these knowledge gaps.

**Abstract:**

Cetaceans have undergone profound sensory adaptations in response to their aquatic environment during evolution. These adaptations are characterised by anatomo-functional changes in the classically defined sensory systems, shaping their neuroanatomy accordingly. This review offers a concise and up-to-date overview of our current understanding of the neuroanatomy associated with cetacean sensory systems. It encompasses a wide spectrum, ranging from the peripheral sensory cells responsible for detecting environmental cues, to the intricate structures within the central nervous system that process and interpret sensory information. Despite considerable progress in this field, numerous knowledge gaps persist, impeding a comprehensive and integrated understanding of their sensory adaptations, and through them, of their sensory perspective. By synthesising recent advances in neuroanatomical research, this review aims to shed light on the intricate sensory alterations that differentiate cetaceans from other mammals and allow them to thrive in the marine environment. Furthermore, it highlights pertinent knowledge gaps and invites future investigations to deepen our understanding of the complex processes in cetacean sensory ecology and anatomy, physiology and pathology in the scope of conservation biology.

## 1. Introduction

Cetaceans, including whales, dolphins, and porpoises, have evolved a complex set of sensory adaptations that enable them to navigate, communicate, and forage in the aquatic environment [1]. These adaptations include specialised structures in their skin, eyes, ears, and nervous system that are modified to detect various physical stimuli, such as sound, light, and pressure [2]. Moreover, cetaceans, like all mammals, rely on a multisensory integration of these stimuli to build a detailed picture of their environment and make decisions based on the information they receive. This process involves a complex interplay between their physical sensory systems and the brain, which is responsible for integrating and processing information from different sensory modalities to form a cohesive understanding of the environment. This complex interplay between peripheral sensory structures and central neural processing systems underscores the remarkable sensory physiology of cetaceans, which remains poorly understood in many aspects. Recent publications have collected current knowledge about the complex anatomy and aspects of the sensory physiology of dolphins [1,3], and brought forth key unresolved questions in the field of sensory ecology [4]. It is of particular importance to understand and combine these contributions to understand sensory functioning under healthy conditions, while we know that cetacean behaviour, physiology, and pathology can be profoundly affected by human disturbances, like noise pollution and habitat destruction. In addition to [5], in which the focus is on perception from the point of view of the bottlenose dolphin (*Tursiops truncatus*), this review focuses on the neuroanatomy involved in the detection of a stimulus, the signal transduction through associated peripheral and central nervous tracts, and the regions of the brain responsible for signal processing. In this paper, we aim to facilitate a better understanding of the interplay of these systems by providing a comprehensive overview of the current state of neuroanatomical knowledge of the sensory systems and recent developments in cetacean sensory neuroanatomy.

## 2. Visual System (Vision)

The cetacean eye has had to readapt to the different conditions that constitute the aquatic medium, including via the shape and thickness of the eye and its components, together with adaptive muscular distribution and vascularization. The eye is fully developed at birth [6]. In order to maintain proper eye and retinal function under changing internal and external pressure conditions, a complex network of vascular tissue (rete mirabile) can assist in eye homeostasis through the adequate distribution of liquids, heat and pressure [7,8,9], which may also be regulated in part by the lamellar mechano-sensors situated in the iridocorneal angle of the eye [10,11]. Overall, the uvea is very well vascularised and the ciliary body is poorly developed, containing nonetheless encapsulated sensory corpuscles in most species, hinting at a certain capacity for aqueous humour regulation and, hence, intraocular pressure [11,12]. Contrary to common belief, eye movements in cetaceans (at least in the bottlenose dolphin) are possible and are also more complex than in most other mammals [13,14,15].

The optic sensory innervation of the toothed whale eye consists of photoreceptors, situated in the retina covering the caudal end of the eye (Figure 1a,b). The retina is thicker than in terrestrial mammals and contains two areas of higher density of retinal ganglion cells [8,16]. The photoreceptor layer in cetaceans (against the large, blue *tapetum lucidum*) consists of rods for 98% and cones for 1–2% [8], both receptive to photons (light energy), although rods are used for low-light-level grayscale vision, and cones for colour vision (Figure 1b). The few cones present in the cetacean retina are long-wavelength sensitive (L-opsin) cones [17], found in most odontocetes, except in some species (including sperm whale, Sowerby’s beaked whale and balaenopterids), but their sensitivity seems to have shifted to blue light, which is still under discussion [18]. Similarly, cetacean rods absorb optimal spectra for their foraging depth [19], to a deep-sea signature of 486 nm, which seems to have been the major selective driver of rhodopsin changes in cetaceans before the odontocete-mysticete divergence [20,21]. Further changes adapting to depth followed in various species [19], especially in deep divers [22]. Melanopsin, a third and separate photoreceptor found in retinal ganglion cells, also appears to have been modified and could improve photobleaching prevention in cetaceans, despite the limited knowledge about its role in this regard [19]. The ganglionic layer of the retina contains relatively few neurons (retinal ganglion cells, RGCs), although with large cell bodies (up to 75–80 µm in diameter) and thicker axons than in terrestrial mammals, which also correspond to a relatively low number of axons in the optic nerve, while still maintaining a consistent overall diameter [8,16,23]. The RGCs form two zones of high density, which correspond to the pinholes formed by the closed peculiar cetacean pupil (Figure 1c) [24]. This specific pattern of the RGCs on the retina has sometimes been associated with a double fovea, although the fovea is a small spot formed by a high density of photoreceptors (cones) and is actually devoid of RGCs. Nonetheless, high-density spots of RGCs on the cetacean retina are indicative of an increased information summation and transmission in the area, which coincides with the shape of the pupil (Figure 1d). It should be noted that this pattern is also similar to the visual streak found in most artiodactyls, corresponding to the shape of the pupil and aligned with the horizon. Recent efforts to map RGCs have been made in several cetacean species [6,23,25,26], but still very few studies have looked at an immunohistochemical characterisation [27].

Famously in 1982 and 1983, Dawson and colleagues reported that the optic nerve of the bottlenose dolphin comprised fibres that were sheathed in more myelin than in any other mammal previously studied [28,29], but ranged only from 200–400,000 in total, which is about a fifth compared to humans. Notable exceptions are the Amazon River dolphin (*Inia geoffrensis*) and Indian river dolphin (*Platanista gangetica*), with around 15,000 fibres. Interestingly, although without subsequent investigations, Dawson and colleagues also noted that a large proportion (about 50%) of the optic nerve cross-section was made of extra-neuronal space, implying the potential importance of the support matrix and cells to the nerve, notably populated by astrocytes [30].

The decussation of the fibres at the optic chiasm reflects an apparent absence of a binocular field and is therefore almost complete [31,32]. Various authors have pondered over the potential binocular nature of vision in dolphins, and their gaze, but no definitive answer has been put forward so far. It seems hard to conceive that a sperm whale may have a meaningful binocular field, and this goes for most mysticetes. Considering the eye anatomy and its possible movements inside the orbit and the neighbouring tissues [15], smaller whales and dolphins may possess a usable binocular field. However, quantitative studies [31,32,33] seem to indicate otherwise. Notably, the eyes have also been reported to move independently [34]. Usually, some fibres at the chiasma terminate in the suprachiasmatic nucleus, attaining the circadian clock, but no such fibres have yet been found in cetaceans [31]. Few records go beyond this step, especially regarding the lateral geniculate nucleus [35,36], rostral (superior) colliculus (SC), and other lower visual centres. The lateral geniculate nucleus (LGN), which is where the retinal ganglion cells synapse, sits far caudo-dorsally from the thalamus, and is reported as laminated only in one case [37] using calcium-binding proteins. The rostral (superior) colliculus receives retinal inputs in the cortex and is a multimodal integration centre, including auditory and somatosensory factors. It has not received systematic attention [38], except for almost a century ago [39,40] and in one instance, where a clear laminar cortex-like structure was described [37].

Visual inputs reaching the visual cortex have also been noted to be only contralateral [31,41]. These electrophysiological experiments have shown that the location of the primary visual cortex (V1) is similarly located to that of sheep or cattle, along the lateral gyrus, more on the dorsal than the occipital vertex; however, its precise location is within the entolateral (endo-marginal) sulcus [42,43,44]. In contrast to terrestrial mammals, the cetacean cortical structure lacks a typical fourth layer, usually made up of granule cells receiving inputs arriving from the LGN. Calcium-binding proteins show a distribution in the would be neighbouring layers [37,45]. In the primary visual cortex, as in other areas of the dolphin brain in general, the current hypothesis is that inputs reach the cortex via the molecular layer and synapse, primarily with extraverted neurons in layer II [1]. The area named secondary visual cortex (V2) in cetaceans [46] is located around (or laterally to, depending on the author) V1 and has a slightly more homogeneous aspect, with a repartition of calcium-binding proteins [42,43], whose arrangement is slightly different from that in other mammals [47]. Potential associative areas, such as V3 and V4, which are not well-defined in most mammals [48], remain to be found in cetaceans. Recent findings using magnetic resonance imaging seem to confirm the absolute vicinity of primary visual and auditory areas [49] to a confounding degree. See Figure 1 for an overview of the neural pathway.

The visual system has undoubtedly received the second most attention, after the auditory system. Nevertheless, various segments along its pathway remain underexplored, with only a limited number of authors describing them. Studies have shed light on the cortex, eye, and optic nerve and tract, but the rest of the intricate pathways require further investigation using modern techniques. Current and future imaging studies, in vivo or ex vivo, will help fill some of these gaps.
Figure 1Drawing of the anatomical components of the visual sensory system. Top. Visual sensory pathway from the optic nerve (II) through the optic chiasm (OC), the location of the rostral (superior) colliculus (SC), the lateral geniculate nucleus (LGN) before reaching the primary (V1) and secondary visual cortex (V2). (**a**) Schematic view of the retinal thickness. Note the large retinal ganglionic cells and the rare cone among rods. (**b**) Drawing of a dolphin eye cut along its longitudinal plane. Note the closed iris and the direction of the light. (**c**) Illustration of the distribution of retinal ganglionic cells (RGC) (blue dots) on the retina (Re). (**d**) View of the matching of the iris opercula and RGC distribution. 1: tapetum lucidum (pigment layer), 2. photoreceptor layer, 3. outer nuclear layer, 4. outer plexiform layer, 5. inner nuclear layer with bipolar, amacroine and horizontal cells, 6. inner plexiform layer, 7. ganglion cell layer, 8. nerve fibre layer, BPC: bipolar cells, CB: ciliary body, Co: cone (blue-shifter L-opsin), Cor: cornea, Ir: iris, Le: lens, LR: light ray, OC: optic chiasm, SC: rostral (superior) colliculus, Ro: rods, RV: retinal vessel, TL: tapetum lucidum, Uv: uvea (choroid), ZF: zonula fibres. Drawn and modified after [1,8,44].
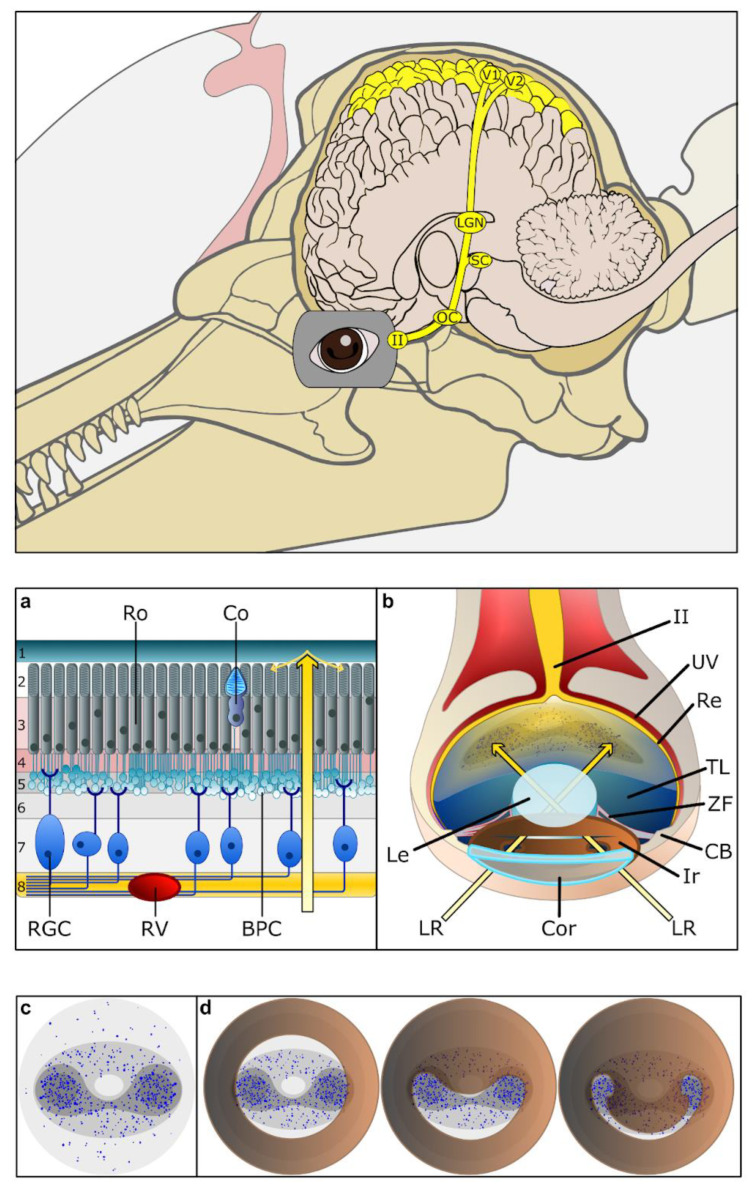


## 3. Auditory System (Hearing)

Cetaceans are highly adapted for underwater hearing (see review of anatomical descriptions of the outer, middle and inner ear of odontocetes and mysticetes in [50]). The external pinnae are absent in cetaceans and the external auditory canal no longer presumably functions as a direct sound conductor between the external environment and middle ear, as cetaceans have developed alternative acoustic pathways [51,52,53,54,55]. The middle ear ossicles are exceptionally dense and stiff [56,57]. The middle and inner ear are enclosed by the tympano-periotic complex (Figure 2), which is suspended by ligaments in the peribulbar cavity in odontocetes, and partially or totally fused to the skull in beaked whales and mysticetes, respectively.

Odontocetes possess a highly developed auditory system, with a large frequency range of hearing (from 100 Hz to over 150 kHz). Most of the knowledge of marine mammal auditory capabilities has been obtained from electrophysiological or behavioural studies [58,59].

The cochleae of cetaceans and terrestrial mammals are largely comparable (Figure 2a–c). They contain the organ of Corti, or hearing organ, which is formed by sensory cells (hair cells) and supporting cells (Figure 2d,e). There are two types of hair cells: the inner hair cells (IHCs), which are responsible for the transduction of the mechanical stimulus into the release of glutamate onto the afferent auditory nerve fibres, and the outer hair cells (OHCs). The OHCs are responsible for the amplification of the incoming sound stimulation within the cochlea and are essential for frequency selectivity and sensitivity. Prestin is the motor protein of OHCs [60]; it is responsible for electromotility (changes in length) and it is required for cochlear amplification. The supporting cells include the inner sulcus cells, border cells, inner and outer pillar cells, three rows of Deiters’ cells, Boettcher cells, Hensen’s cells and Claudius cells. The stereocilia of hair cells are in contact with the undersurface of the tectorial membrane while most supporting cell types are in contact with the basilar membrane.

When a sound wave is transmitted throughout the cochlea, the basilar membrane vibrates, with a maximum of vibration depending on the incoming frequency [61]. While the high-frequency sounds are encoded in the base (closest to the stapes), the low frequencies are encoded at the apex or tip of the spiral [62]. With the vibration of the basilar membrane and the amplification of the sound stimulation by the OHCs, the stereocilia of IHCs bend in contact with the adjacent tectorial membrane. Deflections of stereocilia induced by sound cause the opening of the mechano-electrical transduction channels, which allows the entrance of ions (mostly K^+^ and Ca^++^) into the hair cells and their depolarization [63,64,65,66].

Detailed descriptions of the cochlear morphology using light microscopy were conducted for bottlenose dolphin and Pacific white-sided dolphin [67,68,69,70], while the cochlear fibres in diverse species [71] and features of the basilar membrane in different odontocete species have been compared and related to their hearing capabilities [72,73,74]. Morphological description of the ultrastructure of the cochlea and associated innervation from several species of odontocetes using electron [75,76,77,78,79] and confocal microscopy [79,80,81] were presented. Unique ultrastructural features of the sensory cells and supporting cells of the organ of Corti of odontocetes are found in the basal turn of the cochlea, where the high frequencies are encoded [76,77]. Among other striking features, the OHCs are extremely short and strongly attached to Deiters’ cells, their cuticular plate is thick, and Deiters’ and pillar cells have a highly developed cytoskeleton. All these adaptations, together with a short and thick basilar membrane [50,67,69,72,73,74,76,77], would allow the whole organ of Corti to vibrate at extremely high rate. These features were previously described in the horseshoe bat (*Rhinolophus rouxi*; [82]), a species of echolocating bats, suggesting a convergent evolution among echolocating taxa.

### 3.1. Innervation of the Cochlea

There are four types of cochlear innervation in mammals (see review from [83]), formed by type I and II afferent innervation, and medial and lateral efferent innervation (Figure 2d).

The depolarization of IHCs results in the release of the neurotransmitter glutamate, onto the type I afferent neurons [84]. These neurons send auditory information from IHCs to the brain. This is the only type of innervation that has been studied and described for some species of odontocetes [50,56,67,70,72,73,74,76,77,85]. The number of spiral ganglion cells (SGCs, cell bodies of type I afferent neurons) in odontocetes are up to three-fold larger than in humans, which have a cochlea of similar length. The high number of type I afferent neurons can be related to the complexity of the information from echolocation signals, allowing better frequency discrimination.

Type II afferent innervation sends information from OHCs into the brain. Type II afferents might convey messages of OHC damage or “sensations of auditory pain” to the brain in response to high-intensity noise [86]. Lateral efferent innervation sends information from the brain (lateral superior olivary nuclei) to the dendrites of type I auditory nerve afferent fibres, beneath IHCs, and may provide protection to the auditory nerve. Medial efferent innervation sends information from the brain (medial superior olivary nuclei) to the OHCs and acts to turn down the gain of cochlear amplification (see review by [87]), while the same pathways may also serve to protect from excessive acoustic stimulation and cochlear neuropathy [88,89]. Medial efferent innervation can also aid the detection of signals in noise [90], which can be modulated by attention. While it is likely that cochlear innervation in cetaceans is comparable with terrestrial mammals, descriptions of type II afferent and efferent innervation patterns in cetaceans are still lacking. These descriptions and potential comparisons with terrestrial mammals are crucial to better understand mechanisms of protection to noise exposure, as well as their potential role in conditioned sound attenuation [91,92,93,94].

The fibres of the bottlenose dolphin auditory nerve are on average two–three times as thick as those of terrestrial mammals, and their diameter can change drastically along their longitudinal course (0.42 to 7.3 μm at a distance of 20 μm), without affecting the thickness of the myelin sheath [95]. Evidence of collateral branching fibres within the auditory nerve has been found, although it is unknown whether these form anastomoses with other auditory, vestibular, or intermediate nerve fibres [96]. Nerve density within the central segment of the auditory nerve in bottlenose dolphins (between the internal auditory meatus and the connection to the ventral cochlear nucleus (VCN) of the brainstem appears to range between three and five thousand fibres/mm^2^ [71,97,98,99]. Jacobs and Jensen (1964) estimated total fibre numbers for the auditory nerve in three mysticete species and the sperm whale (*Physeter macrocephalus*), noting the latter’s higher fibre number and diameter [100]. Cetaceans are claimed to have a similar cochlear to vestibular fibre ratio (roughly 60% cochlear to 40% vestibular) of the vestibulocochlear nerve as humans [101]. Too few specimens have been examined to determine if there is any left–right asymmetry.

### 3.2. Central Auditory Pathways

Previous work has described the histology and central nervous auditory pathways of several cetacean—predominantly odontocete—species [102,103,104,105,106,107,108,109,110,111,112,113], describing neuronal morphology and approximate fibre orientation. Overall, the ascending auditory pathways display many similarities with that of other mammals in terms of the grey matter nuclei involved. The cochlear nucleus receives signals from the cochlear nerve and relays them to the trapezoid body, ipsi- and contralateral superior olivary nuclei, lateral lemniscus, inferior colliculus and its brachium, the thalamic medial geniculate nucleus, and finally radiating fibres to the cortices [103] (Figure 1). However, the authors mentioned above note some differences from humans and among the few cetacean species examined. For instance, Breathnach (1960) observed that Monakow’s *striae acousticae*, which connect the posterior cochlear nucleus neurons with the lateral lemniscus and the central nucleus of the inferior colliculus, are not apparent, while Held’s striae (connecting the VCN and bilateral ventral nuclei of the trapezoid body, superior olivary and perio-livary nuclei) [114] are clearly visible. More recent studies specify morphological cell types (e.g., spherical neurons in the VCN or principal neurons in the lateral superior olive) comparative to terrestrial mammals [115] and provide estimates for the number and density of neurons, and the volume of the VCN and the lateral superior olive [116,117]. See Figure 1 for an overview of the auditory pathways.

Several authors have attempted to derive the characteristics of auditory signal processing using proxies like modelling interaural time differences [118] or morphological details, such as the slanted cell arrangement of neurons in the VCN, medial nucleus of the trapezoid body, and the ventral nucleus of the lateral lemniscus, allowing for a millisecond-scale temporal differentiation of signal reception between the more dorsal and more ventral cell rows [119,120]. Another example of this connects the strong structural reduction of the dorsal cochlear nucleus, which is associated with the motor control of external pinnae in terrestrial vertebrates, to an evolutionary loss of function in cetacean species that lack ear adnexa [112].

There is no clear correspondence between cyto-architectural studies and electrophysiological tests, which suggest that the midbrain is the main site of evoked potentials for echolocation clicks, while the posterior lateral temporal cortex responds to lower-frequency sounds [121].

Little is known about the relative quantity of neuronal subtypes for other nuclei of the central nervous auditory pathway. Qualitative, comparative analyses between the nucleus of the lateral lemniscus of bottlenose and striped dolphins (*Stenella coeruleoalba*) and echolocating bat species reveal similar densely packed, ovoidal and multipolar cells, larger in the dolphins (15–25 μm diameter), although the clear columnar arrangement seen in the anterior VCN and medial nucleus of the trapezoid body was less prominent here [122]. Thick and highly myelinated fibres ensure high brainstem transmission speeds exceeding that of terrestrial mammals, despite having to traverse relatively longer distances [123]. 

The caudal (inferior) colliculus (IC) is a mesencephalic nucleus that is between 12 times larger in volume than its human counterpart [124], being relatively larger in odontocetes than in mysticetes, along with other auditory central nervous nuclei [125]. It is a major integration centre for both auditory and non-auditory signals, and one of the most metabolically active areas of the dolphin brain [126]. While a basic division between a fibre-rich tectosome and a (predominantly multipolar) neuron-rich central nucleus is evident [117], subdivisions of these two areas are debated [102,127]. 

The MGN is one of the most uniform segments of the thalamus in dolphins, characterised by small, rounded neurons hugging the brachium of the IC. This small-celled division lies over a magnocellular ventral sector [35]. 

Auditory evoked potential studies place the primary auditory cortex (A1) in the supra-sylvian gyrus, with secondary auditory cortical fields (A2) lateral to it in the ecto-sylvian gyrus [46,128]. These areas, and an auditory receptive field in the temporal cortex, have been seen in tracing studies using dyes, auditory evoked potentials, and ethically unrepeatable experiments with direct access to the brain [129] and have been revisited using cyto-architectural studies [130,131,132,133] and post-mortem magnetic resonance imaging (MRI) techniques [49,117,133,134]. A1 is the thickest out of the main cortical fields in cetaceans [135]. Stereological assessments show that neural density in layers III and IV of A1 areas tends to be higher than in motor and somatosensory neocortical fields, and in the bottlenose dolphin it is on par with the density of V1 [136,137], but the average neuronal size is slightly smaller. Slender, bipolar neurons characterise this cortex, and well-developed cortical columns are apparent in the middle rostro-caudal cortical segment, though this structure is less evident in more rostral and more caudal segments of the parietal cortex [1,132]. Moreover, layer III was found to be generally denser than layer V in all areas studied, suggesting an increased connectivity of that layer [136]. Species differences in the extent of auditory cortices are evident among odontocetes, with *Platanista* species displaying a relatively larger and more densely packed A1 compared to bottlenose dolphins or harbour porpoises (*Phocoena phocoena*), in addition to having a more restricted V1 with a lower neuronal density [138].

Semi-quantitative estimates of auditory pathway connectivity have been put forth using diffusion tensor imaging and preliminary findings suggest auditory–visual integration on both the cortical and subcortical levels (especially in the midbrain) [49,99]. However, further studies linking morphology, connectome, and quantitative parameters are important to expand the current understanding of cetacean auditory perception and its integration with higher cognitive functions.
Figure 2Drawings of the anatomical components of the auditory sensory system. Top. Auditory sensory pathway from the inner ear to the auditory cortex. (**a**) Drawing of the cochlear spiral connecting round (RW) and oval (OW) windows. Note the 2.5 spiralling turns. (**b**) Drawing of a transverse section through the cochlea showing the vestibular (VS), cochlear (CS) and tympanic scala (TS), and the auditory nerve (AN) exiting the modiolus. (**c**) Detail of the inset in (**c**) showing the three scala and the central organ of Corti (Inset frame) and the spiral ganglion (SG). (**d**) Schematic drawing of the structure of the organ of Corti. (**e**) Top view of the reticular lamina after removal of the tectorial membrane (TM), showing the organisation of the hair cells (IHC, OHC), and the supporting cells. 1. Ventral cochlear nucleus and small dorsal cochlear nucleus, 2. trapezoid body, 3. superior olivary complex, 4. lateral lemniscus, 5. caudal (inferior) colliculus, 6. brachium of IC, 7. medial geniculate nucleus, 8. primary auditory cortex, 9. secondary auditory cortex, 10. auditory cortex in temporal lobe, BM: basilar membrane, DC: Deiters’ cells, IHC: inner hair cell, IP: inner pillar cell, LE: lateral efferent innervation, ME: medial efferent innervation, OHC: outer hair cell, OP: outer pillar cell, Pe. periotic bone, TI: type I afferent innervation, TII: type II afferent innervation, Ty. tympanic bone. Drawn and modified after [1,75,76].
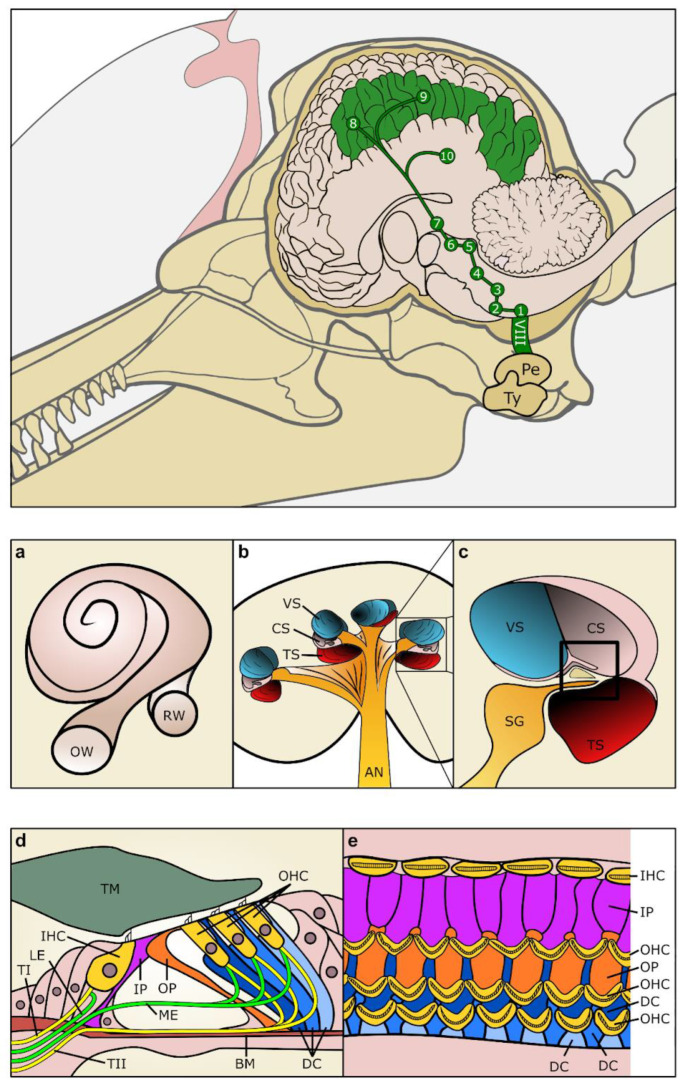


## 4. Vestibular System (Spatial Orientation)

The cetacean vestibular system consists of typical semicircular canals and a membranaceous otolith system [139]. The semicircular canals in cetaceans are extremely small and were considered incomplete and possibly vestigial [140], although this was later proven otherwise [141]. The canals are relatively smaller in odontocetes than in mysticetes when compared to body mass [96,141], with the lateral semicircular canal being relatively larger. In both odontocetes and mysticetes, the three *cristae ampullaris*, as well as the maculae of the utricle and saccule, are developed and covered by a sensory epithelium with hair cells that are connected to the nerve fibrils that form the vestibular part of the vestibulocochlear nerve (VIII) [96]. This part of the nerve takes up about 10% [107], or even as low as 5%, [140] of the nerve area. The total number of fibres was reported in bottlenose dolphin, common dolphin, and other small cetaceans to be about 3500–4000 large fibres [142], which is several times less than that of terrestrial mammals. The size of the vestibular nerve is also deemed thin in mysticetes [101]. The amount of vestibular ganglion cells is also several times lower compared to terrestrial mammals, and the overall vestibular sensitivity has been estimated to be about 10% compared to that of mammals of similar size.

In the brain, four vestibular nuclei are commonly described (medial, superior, descending and lateral). While all these nuclei are small, the lateral nucleus (also named Deiters’ nucleus) is the most developed in cetaceans [102,103,143,144]. The medial longitudinal fasciculus carries ascending and descending parts of the vestibular nuclei. Since the functions of the lateral nucleus (and others) include the eye (via the oculomotor, abducens and trochlear nerves) muscle adjustments (ascending fibres) and bodily posture adjustments (descending fibres), one has to consider that these functions bear importance to cetaceans, in particular regarding the importance of Deiters’ nucleus in locomotion, integrated in the cerebellum [143]. On the whole, very scarce work can be found on the vestibular system in cetaceans. More ought to be carried out, considering the potential interesting comparative points of reference.

## 5. Somatosensory System (Somatosensation)

The cetacean skin forms a protective layer against the external environment that is part of the complex somatosensory system, whose functions are not yet fully clear. The interaction between the environment and skin constitutes a combination of mechanical, temperature or chemical types of stimuli. Like all mammals, toothed whales feature a variety of neuroanatomical configurations associated with the somatosensory system. Bauer and colleagues [145] provided an overview of the current knowledge on neurobiology and sensory ecology of the tactile senses in marine mammals, in which they highlighted the lack of knowledge regarding neuroanatomy and sensory physiology. The sensory cells in the skin and annexes that have been described in cetaceans include free nerve endings and lamellar corpuscles, in the skin of the trunk, flippers and fluke, the blowhole, snout, and around the anus and genital slit [146,147,148,149,150]. The corpuscles are also present in continuations of the skin, such as the nasal sac system [151], the external ear canal [152], the irido-corneal angle of the eye, and various other locations, from close to the epidermis to deep in the dermis. These lamellar corpuscles have also been described in spatial association with glandular structures in the cetacean skin and adnexa. As such, there are corpuscles in the stromal tissue of the circumorbital glandular region of bowhead whale (*Balaena mysticetus*) [153], in the subepithelial tissue in the cervical gill slit of the pygmy sperm whale (*K. breviceps*) [154], in the vicinity of the nasal glands in dolphins [151], and in the stromal tissue among the ear canal glands [152].

Other sensory nerve formations (SNFs) have been described in the vibrissal crypts, including indications of Meissner corpuscles in the skin close to the vibrissal crypt, lamellar corpuscles, Merkel cell-neurite (MCN) complexes, and lanceolate endings (Figure 3a,b) [155,156,157,158]. MCN complexes have also been mentioned in the epidermis of beluga whale (*Delphinapterus leucas*), particularly abundant around the blowhole and anus [159]. Similarly, there are mentions of complex axon terminals attached to dermal papillae [146,160], and recently, immunohistochemical techniques have shown the presence of MCN complexes in striped dolphin skin, anterior to the dorsal fin, in clusters at the base of epidermal rete ridges [161]. No other SNFs, sensitive to other modalities, such as thermoreceptors sensitive to heat and cold, which are assumed unlikely to be of major importance for cetaceans, have been described to our knowledge [2].

The skin innervation in both toothed and baleen whales consists of a dense innervation of dermis and epidermis, with a deep dermal/cutaneous neural plexus of relatively large nerve fibres, from which interlaced smaller nerve fibres radiate into the more superficial subepidermal neural plexus, from which they emanate into dermal papillae and epidermal rete pegs to end in the SNFs described above (Figure 3c) [149,162]. Specifically, and differently from terrestrial mammal nerve branching patterns, the axons tend to travel in bundles up to the superficial dermis, rather than displaying the typical branching of individual fibres (85–120 µm in diameter [159]) [162]. Specific to baleen whale somatosensory innervation, there are only a few records. The skin is a sensitive structure that has shown low-threshold mechano-sensory Aβ neural fibres, which have been noted to end at the dermo–epidermal junction basal cells, morphologically similar to the innervation of MCN-complexes, or as rare end bulb or Meissner-like neural structures in the dermal papillae [162]. Older descriptions in mysticetes reported ‘sensory tubercles’, considered to be modified Golgi–Mazzoni corpuscles on the lips and in the oral cavity of rorquals [163].

The innervation patterns are various and complex, including neural bridging between epidermal pegs, and various types of branching and occasional merging in the papillary dermis, often creating an interconnected neural web across dermal papillae [162]. In general, the structure of individual nerve fibres is similar to those of terrestrial mammals, often embedded in a thick perineural sheath, and with indications of a distinct acellular gap between the axonal core and perineural sheaths, possibly as an adaptation to withstand ambient pressure [162]. Such an acellular space was also noted in nerves and lamellar corpuscles in the ear canal of several toothed whales, although, in that case, this was suspected to be an artefact associated with tissue conservation and/or processing [152]. The nerves are often seen accompanying blood vessels, especially in flippers and on the head, a feature they linked to proper temperature homeostasis for function in areas devoid of blubber [159]. Moreover, peripheral nerve fibres in the floor of the oral cavity of rorquals are highly folded and therefore very stretchy in order to be able to accommodate the large volumes of water during lunge and filter-feeding [164,165,166].

All these findings show that the cetacean skin is well innervated and this lends it to be very sensitive to mechanical deformation [167]. Experimental data ([168] extended by [167]) show very intense sensitivity around the lips and eye, then the head and snout, but a much weaker response from the rest of the body. Most sensitive regions seem to be situated around the eyes, in the mouth corners, and around the blowhole, but also the ‘lips’ of the snout and the ‘area of the melon’. Palmer and Weddel (1964) noted a higher density of nerve endings in the snout and around the nipple than in humans. Other areas, such as the jaw crypts, show an intense anatomical innervation, although the area seems less sensitive, at least in bottlenose dolphins [169]. Electrodermal responses in *Delphinus delphis* show the best sensitivity around the blowhole and under the eyes (10 mg/mm^2^), followed by the lower jaw and melon (10–20 mg/mm^2^), and the skin of the back is the least sensitive (20–40 mg/mm^2^) [170], all of which are in the range of high sensitivity in humans (lips, fingertips, eyelids). Although very few reports exist on the somatosensory innervation in mysticetes, these have been thought to suggest a rather high sensitivity in and around the mouth [107,159,171]. In grey whales (*Eschrichtius robustus*), the head regions with vibrissae have shown significantly more innervation compared to the ventral throat region [155]. 

As mentioned above, sensory information for the head mainly passes through the thick trigeminal (V) nerve to the trigeminal sensory root nuclei (spinal, mesencephalic and principal), and the thalamus (ventral posteromedial nucleus), to be further processed in the somatosensory cortex (Figure 3). The trigeminus is the second largest nerve, after the vestibulocochlear nerve (VIII) in odontocetes, and the largest in mysticetes [102,107], and has the highest number of axons, at least in bottlenose dolphin, likely in part due to the fine peripheral sensation associated with sound production feedback loops [32,102]. The trigeminal complex has raised conflicting reports [107]. It is regarded by some as small, compared to that of the elephant, considering the size of the head in whales [103], and well-developed by others [171]. The spinal root nucleus of the trigeminal nerve varies in shape according to the species and contains a medial subnucleus in baleen whales. It is associated in mammals with coarse tactile stimuli, pain and temperature sensations, and also receives afferents from the vagus (X) and glossopharyngeal (IX) nerves. The principal nucleus is relatively developed, more so in mysticetes, likely owing to its discriminative tactile processing of the face and oronasal cavity [103]. A separate cell group was also described dorsal to the principal nucleus, with no homologue in other taxa [103]. The trigeminothalamic tract (trigeminal lemniscus) connects the trigeminal nuclei and the ventral posteromedial thalamic nucleus, its dorsal part being particularly developed, as in ungulates and elephants [171], but is overall relatively small [103].

The electric signals of bodily sensations travel through the responsible peripheral nerves (and ganglia) to the dorsal spinal root and the spinal cord. The signals ascend through the medial lemniscus, up to the thalamus in the brain (ventral posterolateral nucleus), to reach the somatosensory cortex (Figure 3).

The dorsal column-medial lemniscus pathway in the spinal cord, i.e., the proprioceptive somatosensory pathway, is similar to that of terrestrial mammals [103]. However, the dorsal white columns (gracilis and cuneate fascicles), and the dorsal grey matter horns, together with the dorsal roots of the spinal nerves are reduced in cetaceans compared to terrestrial mammals. This is also reflected in a thin caudal part of the medial lemniscus (arising from gracile and cuneate nuclei) and smaller representations of the skin sensation in the somatosensory cortex, and is hypothesised to be associated with reduced skin surface compared to body size and the loss of extremities [1,101]. Pain and temperature sensations are carried out by the spinothalamic pathway, which is also reduced in cetaceans, the dorsal grey horn and *substantia gelatinosa* in particular [103]. Both somatic and visceral inputs (through vagus nerve fibres for the latter) reach the grey matter dorsal horns. Overall, the whole ventrobasal complex of the thalamus, receiving both body and face inputs, seems to be reduced compared to other mammals [36], the medial (face) part being noticeably larger.

The dolphin and harbour porpoise somatosensory cortex is located laterally to the cruciate sulcus, posterior to the pre-cruciate sulcus [168], and extends into anterior parts of the lateral and supra-sylvian gyri (posterior cruciate and coronary gyri) (Figure 3) [128]. It is situated laterally to the motor cortex and separated from it by the cruciate sulcus [168,172]. Details such as the division into a primary somatosensory cortex (S1) and a secondary (S2, located in the insula in humans), or the complete somatotopy maps, are currently unknown. The cortical column has been studied by some [135,136,173,174], confirming the rather homogeneous aspect of sensitive cortices in cetaceans, with no continuous layer IV, and larger cells, especially in layer V.

Regarding its functional properties as a sense organ, the skin is regarded as both a tactile and a kinesthetic organ (i.e., perception of body movements) [1,146], and a low-threshold mechano-sensory system for hydrodynamic stimuli [162,175]. Cetaceans, like all aquatic tetrapods, are likely to use mechanoreception as a hydrodynamics sense [176], such as for water surface interaction at the blowhole [147], or to detect midwater hydrodynamic stimuli in the water column [177]. Research on the cetacean somatosensory system has been largely driven by an effort to understand hydrodynamic processes to minimise turbulence, often for biomimetic reasons [169,178,179,180,181]. However, the skin is also of importance in other contexts, for example in social behaviour through touch [182]. It is even hypothesised that the skin could be used in tacto-acoustic communication, capable of detecting acoustic signals [168,183], and even used to sense hydrodynamic stimuli from prey (in both odontocetes and mysticetes) [145].

### 5.1. Continuations of the Skin

#### 5.1.1. Nasal Sac System

Khomenko (1974, in: [147,151]) described three types of sensory innervation inside the nasal sac system: free nerve endings in spatial association with the connective tissue, vascular structures, glands, and epithelium, and functional involvement in the regulation of local blood circulation and common internal tissue metabolism functions. The second group comprises fibrillar discs in the muscle tissues, associated with muscle activity during respiration. The third group encompasses two variations of encapsulated, lamellated nerve endings, which were identified as Krause’s end-bulbs and Golgi–Mazzoni corpuscles, mainly present in high concentrations in the skin of the blowhole, and even more specifically in the anterior lip, in the dermal part of the papillary layer [147]. (See Table 1 for a comparative list of cetacean SNFs).

Other studies using systematic histological investigations did not identify any sensory epithelium in the nasal complex of harbour porpoise and striped dolphin, but only lamellar corpuscles located subepithelially (sometimes in the papillary layer) throughout the nasal sac system, but in higher numbers, such as in the ventral part of the nasal plugs or caudal to the nasal ligament [151,184]. These were divided into two groups: small, subepithelial lamellar corpuscles situated between the phonic lips (a.k.a. monkey lips) and respective elliptical adipose bodies (a.k.a. dorsal bursae), and larger corpuscles in the subepithelial tissue in the region of the lips, both dorsal and caudal to the anterior and posterior lip [184]. In general, the nasal system sensory innervation is taken care of by the infraorbital nerve, as a branch of the maxillary nerve (V2) of the trigeminal (V), although with a marked asymmetry, with the right side being more developed, at least in striped dolphin [151].

There is uncertainty regarding the terminal nerve in dolphins, and its relation to the nasal sac system and the loss of olfactory capabilities in toothed whales. Unlike terrestrial mammals, which keep olfactory neurons while reducing terminal neurons during ontogenesis, terminal neurons are kept during dolphin ontogenesis. It is hypothesised that the terminal nerve could be responsible for the innervation of mucosal and submucosal tissue in the odontocete nasal sac system, possibly as an autonomic component for glands and blood vessels that can influence the conditions of the nasal sac system and the sound production processes [1,185] (Figure 1). Moreover, the nasal sac somatosensory system is considered mainly pressure-sensitive, and of rapid adaptation, and therefore considered sensitive to ‘acceleration’ or changes in tissue position or relation, which is essential for fine control through neural feedback loops with the motor innervation through the facial nerve for sound production physiology and homeostasis.

#### 5.1.2. External Ear Canal

The external ear canal, although minute in size, is lined with dermal and epidermal tissues as a continuation of the skin. It is innervated with an extensive intramural nervous plexus with the formation of long and convoluted lamellar corpuscles along the entire length of the canal, from the skin opening to the tympanic membrane. The corpuscles are distributed around the meatus in the superficial half of the ear canal, while in deeper tissues, they are concentrated under a convex epithelium, labelled a ‘nervous tissue ridge’ protruding into the ear canal lumen, opposite the cartilage. There are also nerve fibrils in the dermis and free nerve endings projecting into the epithelium. The corpuscles have been described in harbour porpoises, sperm whales, minke whales, and several species of beaked whales and dolphins [152,186,187].

The cranial innervation of the external ear canal is still poorly studied. In dolphins, it is mainly innervated by the mandibular branch of the trigeminal nerve (V3), through the auriculotemporal nerve and several of its branches, although it likely follows the terrestrial mammals Bauplan with secondary contribution through the facial (VII), glossopharyngeal (IX) and vagal nerves (X), and possibly also the accessory nerve (XI) and nerves connected to the most cranial cervical dorsal root ganglia [188]. In delphinoids, V3 exists in the basicranium separately, through the foramen ovale along the posteromedial margin of the alisphenoid bone [189,190,191,192]. It gives off the small auriculotemporal nerve, which turns ventro-laterally in the direction of the tympanic bulla and dorsally near the mandibular joint, where it gives off several thin branches that project into the area of the external ear canal [193]. The auriculotemporal nerve is also described in Risso’s dolphin as a 3 mm thick nerve that branches from the mandibular nerve and runs in a caudoventral direction towards the tympanic bulla, where it gives off small branches [194]. Based on anatomical findings, the external ear canal is hypothesised to function as an independent mechano-sensory organ that could function as a barometer for ambient pressure, as a complementary organ sensitive to low-frequency acoustic signals, or even for proprioceptive qualities, like balance and orientation [152,186].

#### 5.1.3. Vibrissae, Crypts, Tubercles

Vibrissae are tactile hairs located on the heads of many mammals, including dolphins and whales. Mysticetes of all ages present vibrissae that are distributed on their head, while in toothed whales most species are born with vibrissae while not all species carry the hairs into adulthood, leaving behind a “follicle-sinus complex” (FSC) with a still intact internal hair shaft or a “vibrissal crypt” (VC), along with complete loss of hair and reduction of the surrounding tissues (Figure 1a,b). 

The majority of adult toothed whales lack facial hairs and display several paired rows of vibrissae only during foetal and early postnatal stages, while only species of river dolphin display intact vibrissal hairs in adult stages [145,156,158,159,195,196,197]. The vibrissae of odontocetes reach full development during the initial phases of postnatal growth [1,157,198].

Mysticetes of all species studied to date possess several dozens of relatively symmetrically arranged vibrissae from the rostrum, along the upper and/or lower jaws, and sometimes around the blowhole, with counts reaching up to 250 in bowhead whales (*Balaena mysticetus*) and displaying variations in arrangement and distribution [177,179,195,199,200,201,202].

Among vibrissae in odontocetes, we consider several anatomical manifestations with distinct functional implications. There are the fully intact vibrissal hairs as manifested in all but some neonates, like beluga (*Delphinapterus leucas*) and narwhal (*Monodon monoceros*) [159], the FSC of which comprises the vibrissal follicle with an intact hair follicle that does or does not exceed the skin surface, while the VC has lost the typical well-developed configuration, with vibrissal hair shaft, dermal hair papilla, blood sinus, root sheaths and capsule, but rather has its lumen filled with corneocytes and keratinous fibres in a fatty gel-like matrix and with an agglomeration of fat cells instead of a dermal papilla [157,197]. Concurrently, the structure of FSCs also varies among species, while all consist of a central hair shaft and follicle (complete or only its internal part) originating from a richly innervated dermal hair papilla and surrounded by inner and outer root sheaths. External to the sheaths is a large blood sinus connected to the infraorbital vasculature, and the entire structure is embedded in a connective tissue capsule [1,146,157,158,203]. The size of the vibrissae varies among odontocete species, as porpoises and dolphins seem to have smaller diameters than pygmy sperm whales and Guiana dolphins [177]. It is, to date, not fully clear what are the exact functional implications of the anatomical differences, and to what extent and in which species the vibrissae present at birth develop into mechano-sensory FSC’s or electrosensitive crypts. It has been shown in two odontocete species, the Guiana dolphin and the bottlenose dolphin, that adult animals are capable of detecting low-magnitude electric fields [157,197,204]. Further standardised studies across different species would be needed to elucidate the functional morphology and its implications for sensory ecology. 

Mysticetes present fully equipped vibrissae that are firmly anchored in the skin throughout their lifetime [159,195,205]. Although the anatomical configuration and distribution vary among species of mysticetes, the humpback whale presents particular nodular elevations on the head, called tubercles. They have a similar configuration as the vibrissae in other whales, with a central vibrissal hair shaft and follicle, root sheaths and blood sinus, and an outer connective tissue capsule consisting of stratified squamous epidermal tissue surrounded by reticulated dermal connective tissue [155,195,198,200,203,206].

The innervation in odontocetes is mostly given by the deep vibrissal nerve, which originates from the infraorbital (branch of the trigeminal) nerve, and which innervates the vibrissae from its base and laterally, branching before or after traversing the capsule, and extending towards the skin surface [158,197,203]. In some species, smaller superficial vibrissal nerves have been observed approaching the follicle’s base but mostly traversing the capsule more superficially [197,203]. In total, there are several hundred nerve fibres per vibrissal crypt/FSC in both the neonate and adult bottlenose dolphin [197]. The SNFs around the hair follicle include MCN complexes, lanceolate endings, and lamellar corpuscles, possibly in all species and all ages, although this is still a topic of discussion (Figure 3a,b) [156,157,158]. 

The innervation in mysticetes is similar to that in odontocetes, with several hundred fibres from the deep vibrissal nerve that penetrate the capsule at the base of the follicle, and ascending fibres that form an interwoven net of nervous tissue around each follicle [159,203,206,207]. The SNFs that have been described to date are all lamellar corpuscles, mostly concentrated at the base of the follicle, but also along the follicle [195,206,207]. The humpback whale tubercles are innervated similarly as in other mysticetes, with the presence of lamellar corpuscles [195,206]. These reports are likely to provide an incomplete image of the types of SNFs present, and immunohistochemical analysis specific to nervous tissue could provide complementary information on the nature of the mysticete vibrissal innervation.

The vibrissal anatomy seems to suggest a change in function between neonates and adults, transforming from displaying mechano-sensory to electro-sensory capacities. To what extent these capacities are mutually exclusive, and in which species, is still a topic of discussion [203]. In any case, vibrissae are suggested to be involved in feeding and foraging activities, possibly involving active touch, detecting hydrodynamic events, and/or bio-electric fields depending on species and age. Further investigations are needed to clarify the interspecific differences and the evolution of the vibrissal configurations from neonates to adults.
Figure 3Drawings of the anatomical components of the somatosensory system. Top. The main somatosensory pathways of the head and body. Middle. Drawings of the structure of the (**a**) generalised vibrissal crypt (VC), (**b**) follicle–sinus complex (FSC), and (**c**) skin at the microscopic level. Bottom. Detailed drawings of several of the sensory nerve formations, including (**d**) a transverse section through a lamellar corpuscle (LC), (**e**) MCN-neurite complex at the border between dermis (De) and epidermis (Ep), (**f**) lanceolate endings (LE) with accompanying Schwann cells (SC) in the subepidermal tissue of the vibrissal crypt, and (**g**) ‘free’ nerve endings (FNE) penetrating the epidermis. 1. sensory nuclei of the trigeminal nerve (somatic: principal, mesencephalic, spinal; visceral: solitary nucleus), 2. gracilis and cuneate fibres, 3m. thalamic ventral posteromedial nucleus, 3l. thalamic ventral posterolateral nucleus, 4. somatosensory cortex, 5. dorsal root ganglia in spinal cord, AFC: agglomeration of fat cells, Ax: central axon, De: dermis, DNP: deep nervous plexus, DP: dermal papillae, DR: dermal ridges, DVN: Deep vibrissal nerve, Ep: epidermis, pf: peripheral somatic fibres, FNE: free nerve endings, io: infraorbital nerves (to melon, upper respiratory tract, upper lip), ia: inferior alveolar nerve (to mandibular teeth), LN: lingual nerve (to tongue), na: auriculotemporal nerve (to external ear canal), L: lumen of vibrissal crypt filled with corneocytes and keratinous fibres, LC: lamellar corpuscle, LE: Lanceolate ending, MCN: Merkel cell-neurite complex, pp: pterygopalatine nerve (to pterygoid sinus and palate), PA: superior alveolar plexus (to maxillary teeth), SC: Schwann cell nuclei, SNP: subepidermal nervous plexus, URT: upper respiratory tract, VII: maxillary branch of the trigeminal nerve, VIII: mandibular branch of the trigeminal nerve, VC: vibrissal crypt, VS: venous sinus. Drawn and modified after [1,152,158,197].
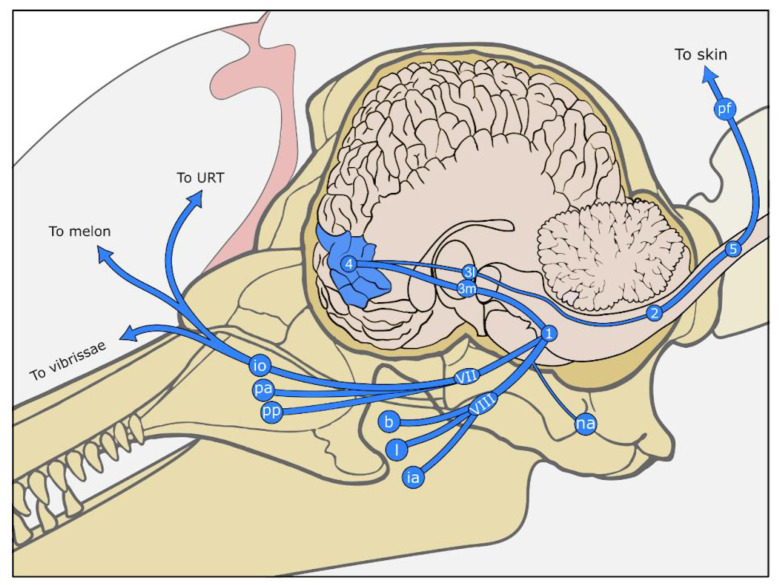

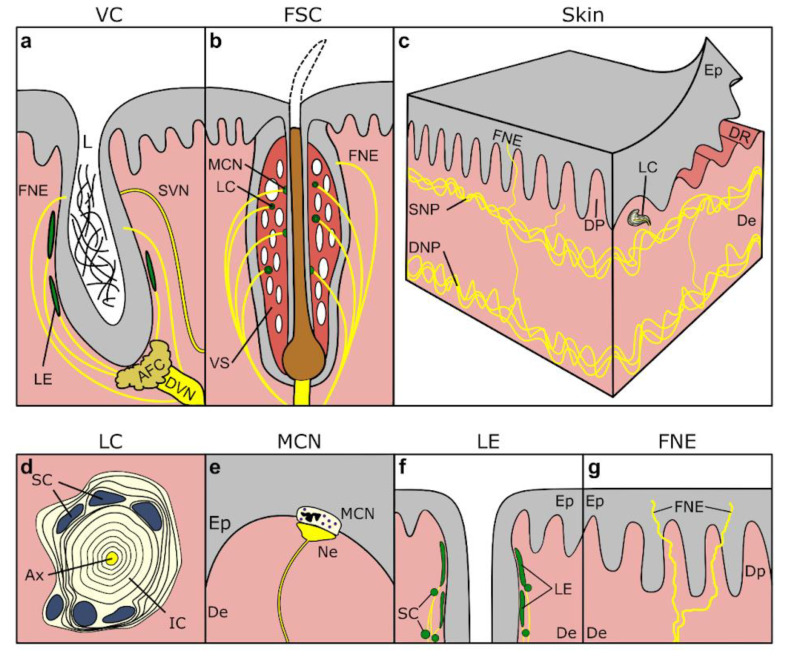

animals-14-00066-t001_Table 1Table 1A non-exhaustive list of the types of SNFs in cetaceans.NameSpeciesLocationReferencesLamellar corpusclesCiliary corpuscles that resemble Pacinian corpuscles*Megaptera novaeangliae*Iridocorneal angle of the eye[208]Corpuscles of Rochon–Duvigneaud*Inia geoffrensis*, *Mesoplodon bidens*, *Ziphius cavirostris*, *Kogia breviceps*, *Delphinapterus leucas*, *Delphinus delphis*, *Tursiops truncatus*, *Stenella attenuata*[10]Encapsulated corpuscles[10]Skin of trunk, flippers and fluke[146]Nasal sac system[151] and Khomenko, 1974 thereinComplex encapsulated terminals, resembling Vater–Pacini corpuscles*Tursiops truncatus*, *Balaenoptera physalus*Skin of the body and head[149]Small Vater–Pacini corpuscles*Balaenoptera physalus*Skin of entire body[209]Encapsulated, lamellated (mechano-) receptor organs*Tursiops truncatus*, *Pseudorca crassidens*Blowhole lips[147](Simple) Lamellar corpuscles*Stenella coeruleoalba*, *Delphinus delphis*, *Tursiops truncatus*, *Ziphius cavirostris*, *Grampus griseus*, *Globicephalus macrorhynchus*, *Berardius bairdii*, *Physeter macrocephalus*, *Kogia breviceps*, *Balaenoptera physalus*, *Balaenoptera acutorostrata*External ear canal[152,186,187]Laminated corpuscles*Balaenoptera spp.*Oral cavity and lipsOgawa Shida, 1950, In: [186]Lamellar corpuscles, similar to Herbst’s corpuscles*B. borealis*Vibrissae[207]Lamellated corpuscles*B. acutorostrata*Ventral pouch[210]Pacinian corpuscles*Balaena mysticetus*, *Delphinapterus leucas*Circumorbital skin[153]Pacchionian bodies*T. truncatus*Mammary glands[159]Krause bulbs*T. truncatus*Flippers, snout, clitoris[159]Golgi–Mazzoni and Pacinian-type lamellated corpuscles*E. robustus*Vibrissae[155]Lamellar bodies/corpuscles*Balaenopteridae**P. blainvillei**T. truncatus*Vibrissae[195][156,157][158]Other SNFLanceolate endings*S. guianensis*, *P. blainvillei**T. truncatus*Vibrissae[156][158]Merkel cell-Neurite Complexes *S. coeruleoalba**S. guianensis*SkinVibrissae[161][157]Intraepithelial ‘free’ nerve endingsAll species studied to dateSkin and adnexa, like vibrissae, external ear canal, etc.E.g., [157,158,162]Intrapapillary myelinated endings (IMEs)*Stenella coeruleoalba*, *Delphinus delphis*, *Tursiops truncatus*External ear canal[152]Muscle spindles15 species of toothed and baleen whalesSkeletal muscle[211,212]Myo-elastic sphincters*Tursiops truncatus*, *Stenella coeruleoalba*Lungs[213]

## 6. Chemoreception

Odontocetes have reduced but functional chemosensory capacities, as shown by anatomical, genetic, and behavioural studies. They are able to detect certain substances in air or water, although the physiology of the sensory systems involved is not well understood [214]. For a brief summary of chemoreception in dolphins, see [5].

### 6.1. Gustation

The sense of taste, known as gustation, plays a crucial role in the sensory experience of various animals. Its significance is particularly noteworthy in cetaceans, given their aquatic environment and the possibility of using taste as a means of identifying food sources. We refer to taste in the strict sense, independent from olfaction and somatosensory perception of the texture, composition, and temperature of an ingested item. The general mammalian anatomical organization consists of taste receptor cells with microvilli, located in taste buds, inside taste papillae on the dorsal side of the tongue and elsewhere in the oral cavity. However, while taste buds are present in foetuses and neonate odontocetes, adults have few or even no papillae and taste buds on their tongue, and discussion of anatomical evidence regarding gustation is ongoing [1,215]. However, in the early years of neonate development, the taste papillae transform into pits that contain microvillar cells at their bottom and are innervated by the glossopharyngeal (IX) and possibly trigeminal (V) nerves, showing connection with the solitary nucleus, a gustatory and olfactory nucleus in the brainstem which projects to the ventro-posteromedial thalamus, overlapping with some olfactory afferents [216,217]. This could explain the behavioural evidence that dolphins can accurately distinguish between chemical stimuli [218].

Regarding mysticetes, to our knowledge, there is no neuroanatomical data on gustation.

### 6.2. Vomeronasal Organ

The vomeronasal organ is considered absent in all cetaceans [2], although there are indications of a vomeronasal-like organ in gray whales [155].

### 6.3. Olfaction

During odontocete ontogenesis, the components of the olfactory system start to develop but rapidly reduce or disappear completely. Adult animals do not possess any olfactory epithelium, olfactory nerve, or olfactory bulb, and the olfactory nucleus and nucleus of the olfactory tract regress to vestigial states in early development [219]. The olfactory tract only persists in the deep-diving sperm whale and bottlenose whale, but its function is not obvious [1,220].

In contrast to odontocetes, mysticetes have a small but functional olfactory system that allows for the detection of airborne odorants [221]. Minke whales (*Balaenoptera acutorostrata*) and bowhead whales (*Balaena mysticetus*) have been shown to possess ethmo-turbinates in the dorsal nasal passage [222,223], and the nasal cavities always contain an amount of air, even when diving [2]. While the nasal gross anatomy has been described in several publications (See [224] for a summary), histological studies and information on the sensory neurons are scarce. In bowhead whales and common minke whales, (mature) olfactory sensory neurons cover at least parts of the ethmo-turbinates, and are similar to those of terrestrial mammals, although they are sparse in comparison [224,225]. The mucous epithelium consists of olfactory sensory neurons, supporting cells, and basal cells, together with the presence of Bowman’s glands. The olfactory nerve bundles, which together form the fasciculated olfactory nerve (cranial nerve I), traverse the cribriform plate to form synapses with the glomeruli of the olfactory bulbs [224,225]. Note that the left and right sides are completely separated [225]. The olfactory bulbs, which relay information from the sensory neurons to higher parts of the brain, are present, but lack glomeruli in the dorsal domain, while the ventral domain contains a higher number of glomeruli than expected from the number of olfactory genes [226,227]. As such, the olfactory nerves enter and connect to the glomeruli in the ventral domain. The size of the olfactory bulb in bowhead whales is about 0.13% of brain mass, which is large compared to the situation in humans at 0.008% [222]. Histological investigations of the olfactory bulb show the presence of all typical mammalian layers, although with specific differences. (See [222] for a detailed description).

Regarding function, humpback whales have been shown to detect krill over a distance of several hundred metres [228], while odontocetes might detect chemicals but over short distances, considering that chemoreception does not play a significant role in foraging behaviour [214]. Besides prey detection, a sense of smell might also be useful for other purposes, such as the detection of conspecifics, or even to extract information on conspecific reproductive states through their breath exhalations [222,229].

In addition, in marine mammals, little attention has been given to the fact that chemoreceptors can also be present in locations other than the nose and mouth [230]. For example, many questions remain regarding the presence of solitary chemoreceptor cells in the respiratory (and digestive) tract in marine mammals, and their role in immune responses [231].

## 7. Magneto-Sensation

Magnetic sensing is the ability to detect the earth’s magnetic field. Cetaceans, and other aquatic mammals, have been described as displaying behaviour associated with orientation, migration, and navigation using magneto-sensation (e.g., [232]), although there is currently no morphological or physiological evidence in cetaceans for any of the known working methods in other animals [233]. One of the methods implies the presence of magnetite particles embedded in mechanoreceptive sensory structures. Magnetite has been described as present in the dura mater of the brain of the common Pacific dolphin, with nerve fibres around the particles [234], and in humpback whale [235], though no particles could be detected in bottlenose dolphins, although this could have been missed due to low-resolution MRI [236]. Another working method was proposed with spontaneously active neurons in an arched trajectory that could be stimulated when electric fields are created when the animal moves through a magnetic field [237].

## 8. Proprioception and Interoception

Proprioception is the sense of body and limb position, movement, and forces, sensed through proprioceptors in muscles, tendons, and joints [238], while interoception refers to the sensation of the internal state of an organism [239]. Documentation of cetacean proprioception is very limited. There are indications of similar SNFs as in terrestrial mammals, considering the structural similarity of the peripheral innervation (e.g., [1,240]). In particular, muscle spindles have been identified in toothed and baleen whale skeletal muscle (longissimus dorsi, masseter) perimysium as small clusters of myo-fibres [1,211,212].

One unique proprioceptive sensory organ of baleen whales, and more specifically of lunge-feeding Balaenopteridae and humpback whales, is the mandibular joint sensory organ situated within the fibrous symphysis of the left and right mandible of individuals of all ages. This consists of “encapsulated nerve termini” and nerves, inside papillae and connective tissue embedded in a gel-like matrix. It is innervated by neurovascular bundles arising from alveolar foramina, through which they are connected to the mandibular nerve (V3) stemming from the trigeminal nerve (V) [241].

### Internal Organs

Interoception is to this day without a precise definition. Afferent sensations of the viscera are essentially mediated by the peripheral somatic nervous system, and unconscious inputs are also mediated by the autonomic (parasympathetic and enteric) nervous system. Visceral afferent inputs are carried to the CNS by neurons in the nodose (and jugular, regarding the auricular and meningeal branches) ganglia from the vagus (X) nerve, or via peripheral somatic fibres in dorsal root ganglia of the spinal cord (Figure 4). The latter run to the dorsal grey horn of the spinal cord and reach the solitary nucleus in the brain stem. Most conscious information regards distention (such as a full bladder), pain and relatively vague sensations. Unconscious inputs are much more specific, such as CO2 levels from the carotid sinus. Visceral activities are often not consciously perceived unless in the case of illness, where it can be projected to somatic surrounding structures.

In cetaceans, there are very few anatomical descriptions of these parts of the nervous system. Existing references (in the harbour porpoise [242] and pygmy sperm whale [243]) show a general organisation similar to that of other mammals. The autonomous system (mostly of the sympathetic part) was described in odontocetes and mysticetes by Agarkov and Veselovsky [244]. The overall importance of autonomic compared to somatic and motor fibres in the fin whale was reported to be small, relative to primates [245]. The cervical sympathetic ganglia are often missing [244]. The hypoglossal nerve detaches in two fine branches to the closely lying vago-sympathetic trunk. The latter separates into vagus nerve and sympathetic chain at the subclavian artery, with the vagus innervating various surrounding vessels, including the aorta, from where it reaches the heart, then the root of the lung, forming a large cardio-pulmonary plexus with several anastomoses. From the caudal end, the left and right vagus nerves continue along the oesophagus. Past the diaphragm, the vagus trunks pass close to the celiac plexus, medial to the adrenals. Fibres from both the vagus and splanchnic nerves also reach the kidneys. The mesenteric ganglia innervation, as pertaining to the enteric system, remains notably under-documented, which aligns with the inherent challenge of their localization.

In mammals, most of the sensitive innervation associated with the heart arises from projected pain (somatosensory). Cardiac autonomic innervation in cetaceans, as described by some, does not differ significantly from other mammals [246]; both vagus nerves give off a branch and sympathetic nerves arise from the middle cervical ganglion. Autonomic sensitivity is therefore likely to reach the nodose ganglion of the vagus, and from there to the solitary nucleus.

The lungs require a little more attention, since physiological processes associated with apnoea are particularly singular in cetaceans. Myo-elastic sphincters at the level of alveolar sacs, with substantial variations among species [213,247], close the airways during a dive. These smooth myo-elastic sphincter contractions are mediated by unmyelinated (autonomic) nerve fibres from the pulmonary plexus. The details of their control are not currently known, and any potential related afferent sensitive input has not been described.

One of the few reports in the literature of central regulation is that of Dell and colleagues describing the orexigenic system in the hypothalamus of the harbour porpoise [248], which appeared to be richly endowed with both parvo- and magnocellular components compared to that of the giraffe, another Cetartiodactyl. Additional information regarding the sleep–wake cycle and neuro-modulatory systems seems to indicate that the evolution of cetaceans has had a profound impact on most systems [249,250,251,252], but a complete integration of these systems into function is lacking.

Sensations associated with the head, such as increased temperature or pain, pass through the trigeminal, facial, glossopharyngeal or vagus nerve to the spinal nucleus of the trigeminal nerve. The central temperature in the brain is directly sensed by the preoptic area [253], which is relatively small in cetaceans. Thermoregulation in general in cetaceans has stirred some debate. Since the aquatic environment is very conductive, the thick blubber layer and a dedicated circulatory system with periarterial veins [1] maintain a fine thermic balance between excessive heat loss in cold waters and overheating in tropical shallow waters. In this context, the vascular structure irrigating the brain holds a degree of mystery regarding its capacity to cool the brain efficiently, since its attributed dampening function linked to diving behaviour necessarily reduces the cooling capacity of blood flow. Work on this pertains more to physiology, but the interested reader can consult [254].
Figure 4Simplified overview of the signal pathways of the various sensory systems in a generic dolphin. Visual system (yellow): Photoreceptors in the eye are innervated by the optic nerve (II), which projects into the optic chiasm (OC), the lateral geniculate nucleus (LGN) (2), rostral (superior) colliculus (SC), primary visual cortex (V1) (4), and secondary visual cortex (V2) (5). Auditory and vestibular system (green): Ear and vestibulum give rise to the cochlear and vestibular parts of the vestibulocochlear nerve (VIII). The cochlear part of the nerve projects to the ventral cochlear nucleus (1) (VCN) (and small DCN), trapezoid body (2), superior olivary complex (3), lateral lemniscus (4), caudal (inferior) colliculus (5) (IC), brachium of IC (6), medial geniculate nucleus (7), primary auditory cortex (8), secondary auditory cortex (9), and auditory cortex (10) in temporal lobe. The vestibular part of the nerve projects to the four vestibular nuclei (1) in the brainstem of which Deiters’ nucleus (lateral nucleus) is of greatest functional importance. Somatosensory system (blue): Peripheral somatosensory structures of the head, including vibrissae and skin and adnexa such as URT and external ear canal, innervated by branches of the trigeminal nerve (VII and VIII). This nerve projects into the sensory nuclei of trigeminal nerve (1) (principal, mesencephalic, and spinal nucleus), from which connections radiate into the ventral posteromedial thalamic nucleus (3m); Somatic innervation from the body goes through peripheral somatic fibres (pf) that connect to dorsal root ganglia in the spinal cord (5), which connect to relatively weak gracilis and cuneate fibres (2) in the brainstem and from there to the ventral posterolateral thalamic nucleus (3l). Both thalamic nuclei project into the somatosensory cortex (4). Interoception (light blue) from the internal organs such as the heart, lungs, gastrointestinal tract and urinary tract, are innervated by branches of the vagus nerve (X). The vagus nerve reaches the nodose ganglion (1) and from there projects into the lateral solitary nucleus (Deiters’ nucleus). The gustatory system (pink) is associated with cells in the oral cavity that are innervated by the hypoglossal nerve (IX), which in turn projects into the solitary nucleus. Redrawn and modified after [1,171,255] and with topographic help from [193,194].
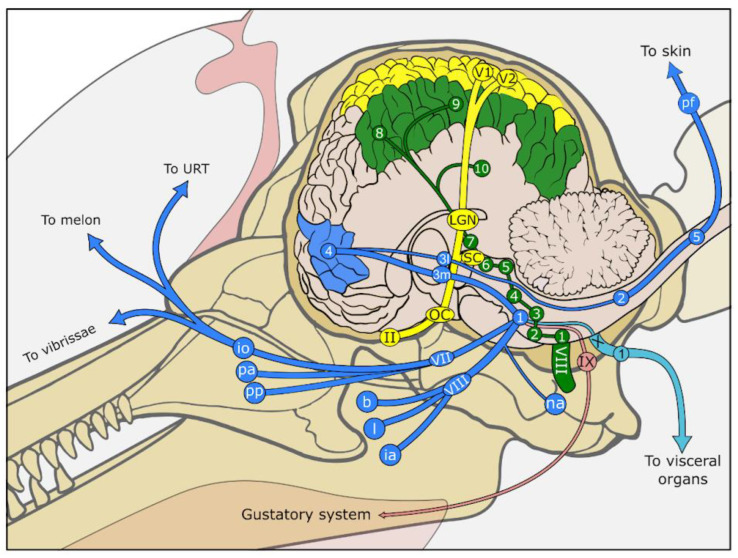


## 9. Central Nervous System Signal Processing

The central nervous system is the primary processing centre for the various sensory systems described above. Regions of the brain are organised hierarchically within it to optimise the integration and interpretation of sensory information. The telencephalon is the neuroanatomical hub where sensory signals, pre-processed by the nuclei of the brainstem and midbrain, are combined into a complete picture of the animal’s environment, and then integrated with other intrinsic signals to eventually create a mosaic of complex cognition. This image interacts with memory, which opens the capacity to learn from past experiences by association with emotional responses. Classically, the prefrontal cortex (PFC) is the host of higher cognitive functions [256], while the limbic system is associated with emotional and instinctive responses, like the activation of the “fight or flight” mode.

Although the general morpho-functional organisation is similar in all mammals, cetaceans show peculiarities concerning sensory information processing in the brain. The somatosensory, visual and auditory primary processing areas have long been discovered, but their limits and connections are still scarcely known (Figure 4) [41,43,47,129,130,168,257,258]. More recent studies have focused on connectivity, especially regarding unmapped territories, notably in the temporal cortex and PFC. Gerussi and colleagues [49] have investigated a putative dolphin PFC in comparison to the human counterpart using tractography, noting that some of the fibre tracts lead via the superior longitudinal fasciculus to the temporal cortex, for which increasing evidence points towards an auditory function [129,133]. This could represent a candidate route for close association in an animal in which auditory input plays a major role in orientation in space and, hence, survival. 

A visual and auditory cross-integration has long been presumed in cetaceans, especially in odontocetes, since they heavily rely on echolocation, and while artefacts, owing to technical limitations of differentiating between auditory and visual crossing fibres within the thalamocortical radiation, deserve due notice, this overlap appears across several studies and merits further investigation [129,133]. At the microscopic level, interneuron labelling has shown that a certain level of additional layering could be found and used to differentiate species and cortical areas [42,45]. A thorough whole-brain investigation remains to be done, both in terms of cytoarchitecture and connectivity, where neuroimaging could serve as a valuable complementary tool [49,259].

An important component of sensory processing and emotional integration resides in the limbic system, the different parts of which are developed to varying degrees in cetaceans. The hippocampus, mammillary body, and fornix are remarkably small compared to other mammals, while the habenulae, anterior thalamic nuclei and especially the amygdaloid complex have notably large dimensions [171,219,260]. This, however, does not seem to impair their mnemonic capacity. Lacking their usual olfactory connections, it is possible that limbic parts interact with auditory signals instead of the visual limbic pathway found in humans, highly relying on vision [261,262]. In line with this is the paralimbic lobe, a unique feature of cetaceans described by Morgane and colleagues [262], covered by a heterotypical sensorimotor-type cortex, and which could constitute a significant area of cross-modality [255]. It differs strongly from the mammalian brain structure, where association cortices tend to separate sensory and motor systems.

The claustrum is one of the basal ganglia that is referred to as the cross-integrator of sensory information, involved in salience processing, with some authors going so far as to call it the “Cartesian theatre” of consciousness (Denneth, 1991, in [263]). It has clear connections to the visual cortex in cats and sheep [264] and has been noted to lack parvalbumin-reactive neurons [265]. However, a clear map of the claustrum’s connectivity is pending investigation. Increased auditory connections would be consistent with the rest of the cetacean brain adaptations.

Another point still containing many open questions is brain asymmetry and lateralization, and how that affects central processing. Although skull asymmetry in odontocetes is prevalent in favour of the right side, brain asymmetry is evident only on a closer look, such as by measuring the hemispheric surface area [266]. Some behavioural evidence for the dominance of the right hemisphere is available in the form of a preference to counterclockwise turning, which appears to be confounded by a slight preference for using the right eye i.e., left hemisphere [267,268,269]. A recent study has examined asymmetry in the white-matter component of the brain and found predominant leftward lateralization, with the notable exception of the rightward fibre tract implicated in communicative function: the arcuate fasciculus [134].

Regarding most of the recent neuroscientific questions, the cetacean sits at an interesting place given its peculiar sensory ecology. Progress in our understanding of their central processing is at a crossroads. A comprehensive atlas of brain connectivity and cytoarchitecture in different cetacean species would be a singular milestone in the study of the evolutionary biology of cetaceans, which are uniquely placed in the evolutionary tree and represent intriguing translational models for the development and function of the human brain.

## 10. Conclusions

Cetacean sensory systems are closely interconnected in their evolution, as the development of one sensory system often results in the reduction of another due to substantial energy demands [270], which is reflected in both peripheral and central neuroanatomical structures. As a case in point, the hearing apparatus in cetaceans exhibits remarkable developments and specialisations at all levels, ranging from the inner ear to cranial nerves and the various stations along the pathway in the central nervous system. Conversely, sensory systems like olfaction and gustation in toothed whales have experienced a clear reduction in their functional anatomy. Furthermore, external stimuli frequently trigger multiple senses, which are integrated within the central nervous system and play a crucial role in complex tasks, such as foraging [271]. 

This neuroanatomical review has taken on a structure according to the classical division of the known sensory systems for clarity, while it underscores the importance of adopting a holistic perspective in studying cetacean sensory systems as complementary to highly specialised investigations. Such integrative approaches can enhance our understanding of these marine mammals’ sensory ecology [270,272,273], for which there remains a substantial gap in basic knowledge of how the sensory systems function. For instance, we lack insights into how these animals may sense and process electric currents, magnetic fields, and information related to hydrodynamic flow, pressure and tension in the skin, the balaenopterids’ mandibular symphysis joint and throat grooves, and other morpho-functional features. All these inputs reaching the CNS form a crucial integrative picture that is poorly represented to date. These mechanisms are essential for their survival in the often extreme conditions they encounter throughout their lifetimes. Furthermore, while expanding our fundamental grasp of sensory anatomy and physiology is crucial, particularly for species conservation efforts [274], it is equally important to consider the ecological significance of various sensory systems. This perspective sheds light on the evolutionary context of each sensory system and its importance within the animal’s overall sensory ecology.

This review ultimately reflects the prevailing knowledge gaps concerning cetaceans, which necessitate further investigation to enhance our comprehension of their sensory neuroanatomy. Our understanding is largely based on seminal research of the previous century, while recent technological advancements, such as neuroimaging, have facilitated substantial progress within specialised domains of sensory neuroanatomy in cetaceans, occasionally yielding preliminary validation of longstanding hypotheses.

In conclusion, we have outlined the fundamentals of cetacean sensory neuroanatomy, from peripheral to central. This knowledge is pivotal to gaining an understanding of how these remarkable animals have adapted to their unique aquatic environment, how they interact with their surroundings and, ultimately, to inform conservation initiatives.

## Data Availability

Not applicable.

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
