# Peer review of "Neuroanatomy of the Cetacean Sensory Systems"

_animals, 2023, doi:10.3390/ani14010066_

Round 1
Reviewer 1 Report
Comments and Suggestions for Authors
This review article is well-written with comprehensive content on neuroanatomy of the cetacean sensory systems. Minor editing of English language is required and here are some of the suggestions:
Line42, finely tuned...perhaps use the term "modified" or "adjusted"?
Line 54, "sensory" functioning
Line 60, In this paper, we are looking to facilitate the understanding of the interplay...bit confused by saying "looking to facilitate"...
Line 70-73, ref shall be in numbering format?
Line 172, Most of the knowledge of marine mam-172 mal auditory capabilities has been obtained from electrophysiological or behavioural 173 studies. - would be good to have a ref on this?
Line 196, The depolarization of IHCs results in the release of the neurotransmitter glutamate, 196 onto the type I afferent neurons [63]. - perhaps could move this down to the section of Innervation of the cochlea?
Line 621, The vibrissal anatomy seems to suggest a change in function between neonates and 621 adults, transforming from displaying mechanosensory to electrosensory capacities. To 622 which regard these capacities are mutually exclusive, and in which species, is still a topic 623 of discussion. - would be good to have a ref on this?
Line 801, Gerussi and his colleagues [47] have investigated ...
Line 813, A thorough whole-brain investigation remains 813 to be done, both in terms of cytoarchitecture and connectivity. - perhaps can add in some studies as ref on using neuroimaging as a tool for such investigation?
e.g. Kot BCW, Tsui HCL, Chung TYT and Lau APY (2020) Postmortem Neuroimaging of Cetacean Brains Using Computed Tomography and Magnetic Resonance Imaging. Front. Mar. Sci. 7:544037. doi: 10.3389/fmars.2020.544037
Line 834, Another point still containing many open questions is brain asymmetry and lateralization....
Line 844, Regarding most of the recent neuroscientific questions, the cetacean sits...
Line 882, while recent technological advancements (such as neuroimaging) have facilitated substantial progress...
Line 899 Ear and vestibulum give rise to...
Minor editing of English language is required and some suggestions had been made accordingly.
Author Response
This review article is well-written with comprehensive content on neuroanatomy of the cetacean sensory systems. Minor editing of English language is required and here are some of the suggestions:
We sincerely acknowledge the comments received and have incorporated all changes. We also have included more illustrations as recommended by both the other reviewer and the journal editor, greatly improving the overall clarity and accessibility of the work.
Line42, finely tuned...perhaps use the term "modified" or "adjusted"?
The term has been changed to ‘modified’
Line 54, "sensory" functioning
This term has been added.
Line 60, In this paper, we are looking to facilitate the understanding of the interplay...bit confused by saying "looking to facilitate"...
The wording has been adjusted. Line 60-63
Line 70-73, ref shall be in numbering format?
The references in the mentioned lines, together with a few other references in the rest of the article, have been changed to numbering format.
Line 172, Most of the knowledge of marine mam-172 mal auditory capabilities has been obtained from electrophysiological or behavioural 173 studies. - would be good to have a ref on this?
References have been added
Line 196, The depolarization of IHCs results in the release of the neurotransmitter glutamate, 196 onto the type I afferent neurons [63]. - perhaps could move this down to the section of Innervation of the cochlea?
The sentence has been moved accordingly and phrasing has been modified slightly. Line 230.
Line 621, The vibrissal anatomy seems to suggest a change in function between neonates and 621 adults, transforming from displaying mechanosensory to electrosensory capacities. To 622 which regard these capacities are mutually exclusive, and in which species, is still a topic 623 of discussion. - would be good to have a ref on this?
The following reference has been added to support this sentence:
Mynett, N.; Mossman, H.L.; Huettner, T.; Grant, R.A. Diversity of Vibrissal Follicle Anatomy in Cetaceans. The Anatomical Record 2022, 305, 609–621, doi:10.1002/ar.24714.
Line 801, Gerussi and his colleagues [47] have investigated ...
We have replaced “Gerussi et al.” to “Gerussi and colleagues”
Line 813, A thorough whole-brain investigation remains 813 to be done, both in terms of cytoarchitecture and connectivity. - perhaps can add in some studies as ref on using neuroimaging as a tool for such investigation?
e.g. Kot BCW, Tsui HCL, Chung TYT and Lau APY (2020) Postmortem Neuroimaging of Cetacean Brains Using Computed Tomography and Magnetic Resonance Imaging. Front. Mar. Sci. 7:544037. doi: 10.3389/fmars.2020.544037
We have added the suggested link to neuroimaging and references. Line 868-870
Line 834, Another point still containing many open questions is brain asymmetry and lateralization....
The sentence has been adjusted accordingly
Line 844, Regarding most of the recent neuroscientific questions, the cetacean sits...
The sentence has been adjusted accordingly
Line 882, while recent technological advancements (such as neuroimaging) have facilitated substantial progress...
The sentence has been adjusted accordingly
Line 899 Ear and vestibulum give rise to...
The verb has been adjusted accordingly
Reviewer 2 Report
Comments and Suggestions for Authors
In this manuscript, the authors realize an extensive review of previous literature dealing with cetacean sensory ecology. The manuscript is well-organized and well-written and can be published after only minor revision.
In general, I appreciate the simple language adopted by the authors that should enhance the readability by readers from different research fields potentially increasing the visibility and citation of this paper. Moreover, I appreciate the depth of the review that really goes deep into the details from both sensory and brain anatomy. I do not appreciate the scarce use of illustrations. There is only one illustration in this review but this decreases the appeal this paper may have and certainly does not help in visualizing at least some of the many anatomical structures discussed in the text. The provided illustration (Fig. 1), moreover, includes some text that is very small and difficult to read so it is not so useful.
Moreover, I do not agree with the authors about the paths followed by the trigeminal nerve to exit the skull and find that in the cranial osteology of both odontocetes and mysticetes a particular foramen (called oval or pseudoval foramen) is present that is not cited in this manuscript. I will focus on this below.
As a whole, I strongly ask the authors to include more illustrations representing at least some selected sensory structures and the more intricate connections to the brain areas involved in sensory ecology. A map of these areas on the cetacean brain would be very useful to help readers to localize the structures the authors are talking about.
Here, a few points for specific corrections and changes.
Line 163: close parenthesis.
Line 173: a list of references is highly desirable here.
Line 175: maybe it is better to write cochleae.
Line 185: a schematic representation of the auditory structures including cell types is highly desirable here because it is hard to follow all the descriptions. In general, I suggest the authors to add more schematic line drawings of pictures of the structures they are describing to enhance the reader’s ability to understand what they are writing. Otherwise this paper will be suitable by specialists only.
Line 197: remove the additional space before [63]
Lines 362-374: it would be very desirable to have the corpuscles illustrated.
Lines 540-543: in Cetacea the trigeminal nerve is not usually associated to the cranial hiatus. In mysticetes, it exits the skull from the foramen ovale (pseudoval foramen of some authors) located within the squamosal or between squamosal and pterygoid, another branch exits from the dorsal infraorbital foramina located on the dorsal surface of the maxilla. It is possible to follow these paths in endocasts of living and fossil cetacean species in which the route through the foramen ovale is clearly indicated (see Bisconti et al. 2021, Journal of Comparative Neurology 529:1198-1227; Bisconti et al. 2022, Brain Behavior Evolution 96:78-90). Anatomical studies by Pilleri in the 80s and 90s clearly showed the distinctness of the root of the trigeminal nerve from the nerve roots exiting the skull from the cranial hiatus. I ask the authors to revise this part accordingly to all these anatomical studies. Probably, the exit from the foramen ovale is what the authors call a fissure between alisphenoid, the periotic and the squamosal. In all the species of cetaceans I know, such a fissure is not related to the periotic.
Line 738: I do not understand the use of italics here.
Figure 1: the text in the colored box is almost unreadable. It is necessary that such a text is wider and better defined.
Author Response
In this manuscript, the authors realize an extensive review of previous literature dealing with cetacean sensory ecology. The manuscript is well-organized and well-written and can be published after only minor revision.
In general, I appreciate the simple language adopted by the authors that should enhance the readability by readers from different research fields potentially increasing the visibility and citation of this paper. Moreover, I appreciate the depth of the review that really goes deep into the details from both sensory and brain anatomy. I do not appreciate the scarce use of illustrations. There is only one illustration in this review but this decreases the appeal this paper may have and certainly does not help in visualizing at least some of the many anatomical structures discussed in the text. The provided illustration (Fig. 1), moreover, includes some text that is very small and difficult to read so it is not so useful.
Moreover, I do not agree with the authors about the paths followed by the trigeminal nerve to exit the skull and find that in the cranial osteology of both odontocetes and mysticetes a particular foramen (called oval or pseudoval foramen) is present that is not cited in this manuscript. I will focus on this below.
As a whole, I strongly ask the authors to include more illustrations representing at least some selected sensory structures and the more intricate connections to the brain areas involved in sensory ecology. A map of these areas on the cetacean brain would be very useful to help readers to localize the structures the authors are talking about.
All co-authors express their sincere appreciation for the comments above and specific points written below. We completely agree with all suggestions and have made diligent effort to incorporate all changes. We also have included more illustrations as suggested, which significantly enhances the overall readability of the work.
Here, a few points for specific corrections and changes.
Line 163: close parenthesis.
The parenthesis was closed
Line 173: a list of references is highly desirable here.
References have been added
Line 175: maybe it is better to write cochleae.
The word has been adjusted as suggested
Line 185: a schematic representation of the auditory structures including cell types is highly desirable here because it is hard to follow all the descriptions. In general, I suggest the authors to add more schematic line drawings of pictures of the structures they are describing to enhance the reader’s ability to understand what they are writing. Otherwise this paper will be suitable by specialists only.
A figure of the auditory structures has been added (Figure 2), together with figures for the visual and somatosensory systems, and a fourth figure as an overview of central nervous system pathways of the various sensory systems.
Line 197: remove the additional space before [63]
The space before the reference has been removed
Lines 362-374: it would be very desirable to have the corpuscles illustrated.
An illustration of the lamellar corpuscles has been added accordingly (Figure 3d).
Lines 540-543: in Cetacea the trigeminal nerve is not usually associated to the cranial hiatus. In mysticetes, it exits the skull from the foramen ovale (pseudoval foramen of some authors) located within the squamosal or between squamosal and pterygoid, another branch exits from the dorsal infraorbital foramina located on the dorsal surface of the maxilla. It is possible to follow these paths in endocasts of living and fossil cetacean species in which the route through the foramen ovale is clearly indicated (see Bisconti et al. 2021, Journal of Comparative Neurology 529:1198-1227; Bisconti et al. 2022, Brain Behavior Evolution 96:78-90). Anatomical studies by Pilleri in the 80s and 90s clearly showed the distinctness of the root of the trigeminal nerve from the nerve roots exiting the skull from the cranial hiatus. I ask the authors to revise this part accordingly to all these anatomical studies. Probably, the exit from the foramen ovale is what the authors call a fissure between alisphenoid, the periotic and the squamosal. In all the species of cetaceans I know, such a fissure is not related to the periotic.
We greatly appreciate this remark as it was an anatomical error on our behalf. Indeed, the trigeminal nerve exits the skull from the foramen ovale. We have changed the text and references. Line 574-576
Line 738: I do not understand the use of italics here.
The use of italics has been removed.
Figure 1: the text in the colored box is almost unreadable. It is necessary that such a text is wider and better defined.
The original illustration has been replaced by four separate figures, as also suggested by the editor. These figures depict the main components of three of the main sensory systems (visual, auditory, and somatosensory), and one figure showing the overview of the central nervous system pathways of the sensory systems mentioned in the manuscript.
Round 2
Reviewer 2 Report
Comments and Suggestions for Authors
Dear Authors and Editor,
This is the second review round that I perform on this manuscript. I find that this version is significantly improved with respect to the first one. In particular, the more accurate description of the trigeminal exit from the skull and the inclusion of more illustrations is the final touch for a well-conceived and well-written piece of science. I feel that this paper well describes the state-of-the-art of our knowledge about cetacean sensory biology and represents a good starting point to go on with new research in this field.
The authors corrected the previous version of the manuscript according to my requests and now I think that the manuscript is acceptable for publication.
Best regards